# Diet Quality and Exhaled Breath Condensate Markers in a Sample of School-Aged Children

**DOI:** 10.3390/children10020263

**Published:** 2023-01-31

**Authors:** Mónica Rodrigues, Francisca de Castro Mendes, Inês Paciência, João Cavaleiro Rufo, Diana Silva, Luís Delgado, André Moreira, Pedro Moreira

**Affiliations:** 1Faculty of Nutrition and Food Sciences, University of Porto, 4150-180 Porto, Portugal; 2Basic and Clinical Immunology, Department of Pathology, Faculty of Medicine, University of Porto, 4200-31 Porto, Portugal; 3Epidemiology Research Unit and Laboratory for Integrative and Translational Research in Population Health, Institute of Public Health, University of Porto, 4050-600 Porto, Portugal; 4Center for Environmental and Respiratory Health Research (CERH), Population Health, University of Oulu, 90014 Oulu, Finland; 5Biocentur Oulu, University of Oulu, 90014 Oulu, Finland; 6Immuno-Allergology Department, Centro Hospitalar São João, 4200-319 Porto, Portugal

**Keywords:** exhaled breath condensate, diet quality, HEI-2015, asthma, sodium, potassium, conductivity

## Abstract

Exhaled breath condensate (EBC) analysis is a recently developed, non-invasive method used to identify and quantify biomarkers, mainly those coming from the lower respiratory tract. It seems that diet can influence the airway’s inflammation and change the exhaled breath composition. This study aimed to assess the association between diet quality intake and markers in EBC among school-aged children. A cross-sectional analysis included 150 children (48.3% females, aged 7–12 years, mean age: 8.7 ± 0.8 years) from 20 schools across Porto, Portugal. We assessed diet quality through the Healthy Eating Index (HEI)-2015, which was estimated based on a single 24 h food recall questionnaire. EBC samples were collected, and we assessed their ionic content (Na^+^, K^+^) and conductivity. The association between diet quality and Na^+^, K^+^, Na^+^/K^+^ ratio and conductivity was estimated using logistic regression models adjusted for potential confounders. After adjustment, a higher quality diet score increases the odds of higher conductivity values of the EBC (aOR = 1.04, 95%CI 1.00; 1.08). Our findings suggest that a higher diet quality in school-aged children is associated with higher conductivity levels of the EBC.

## 1. Introduction

Exhaled breath condensate (EBC) analysis is a non-invasive method used to detect biomarkers primarily from the lower respiratory tract [1,2]. Non-invasive assessment of airways’ inflammation is critical in children since most current techniques assessing inflammation in respiratory diseases are invasive and thus unsuitable for routine use [1].

EBC is a byproduct of the cooling and condensation of the exhaled aerosol from quiet breathing [1,2]. It mainly contains water vapor and a small fraction of volatile and non-volatile macromolecules [3,4]. The varying-sized droplets or particles are considered to be aerosolized from the epithelial lining fluid (ELF) and to reflect the fluid itself [3,4].

This thin fluid layer covering the epithelial surface of the respiratory tract is critical for normal lung function [5,6]. Even among healthy individuals, ELF is slightly acidic, which may be part of normal airway defence. However, acidification of the airway surfaces may be a measure of airway disease, such as asthma, cystic fibrosis, and chronic obstructive broncho-pulmonary disease [7,8].

The ionic equilibrium in the lung is crucial for ELF balance [7,8] and a high-salt diet is associated with an altered cellular Na^+^ homeostasis, which appears to be caused by an inhibition of the Na^+^/K^+^ pump, resulting in a Na^+^ accumulation within airway smooth muscle cells [9,10]. Altered intracellular Na^+^ homeostasis has been linked to allergic asthma [10].

In a study with healthy individuals and subjects with various respiratory diseases, the correlation between conductivity and ammonium was almost perfect. As such, it can be assumed that electrical conductivity is a marker of ammonium content in EBC that is easy to measure and readily available [11]. Acidification of the airways can occur as a result of an increase in free protons or a decrease in alkaline-buffering capacity [12]. Ammonium is a possible ELF buffer and, consequently, ammonium production by the enzyme glutaminase, that generates ammonium through glutamine, could be a significant buffer source determining ELF pH [12,13].

Diet has been associated with asthma pathophysiology [14,15] and has also been able to alter the composition of the exhaled breath [16,17,18,19,20,21,22,23]. In fact, diet may modulate asthma-related miRNAs of the EBC, as school-aged children with a high dietary acid load, most likely consuming fewer vegetables and fruits and more animal protein and animal-based products, had significantly increased miR-133a-3p in their airways [24].

Some exhaled breath volatile organic compounds (VOCs) appear to be influenced by dietary components [16,17,18,19]. Researchers have found that ammonia levels in the breath rise after protein intake [20,21] and that a diet rich in dietary fiber also changes VOC’s levels [22]. A study comparing individuals who commonly consume certain foods (more or equal to the median of the yearly food consumption frequency) with individuals who do not commonly consume the same products (less than median of the yearly food consumption frequency) concluded, by calculating the rate of yearly consumption of individual foods, that individuals that have a common consumption of pork, coffee, legumes, leeks, kefir or fermented milk, and garlic have significantly different concentrations of some VOCs [17]. Furthermore, patients with a routine renal diet (lower in fruits and vegetables and consequently in K^+^) had significantly lower exhaled methanol levels than controls with unrestricted diets and patients on a high-fiber diet [19]. Another study has demonstrated a change in EBC VOCs in healthy individuals on a gluten-free diet for four weeks, suggesting that dietary changes impact exhaled metabolites [18]. Also, the exhaled breath composition of volunteers following vegan and Mediterranean omnivorous diet was different, with a clear separation between some compounds (methanol, 1-propanol, pentane, hexane, and hexanal) [16].

Total EBC NO_2_^−^/NO_3_^−^ levels, biomarkers of the nitrosative pathway and indirect indicators of NO synthesis, are negatively associated with cured meat ingestion but not with leafy vegetable consumption, cured meats being one of the major sources of NO_2_. Conversely, leafy vegetables are amongst the major sources of NO_3_ [23].

EBC contains markers that can be easily and quickly assessed, such as ionic content (Na^+^, K^+^) [25] and conductivity [11], but their relation to diet is still unexplored. As foods are consumed as complex combinations of bioactive components and nutrients that interact with each other and promote synergist effects influencing metabolic and health effects [26,27], recent efforts have concentrated on researching well-validated indices that assess the quality or properties of entire diets [27,28].

Therefore, this study aims to explore the association between diet quality using the Healthy Eating Index (HEI)-2015 and exhaled breath condensate markers (Na^+^, K^+^, Na^+^/K^+^ ratio and conductivity) in school-aged children.

## 2. Materials and Methods

The population of the present study was collected from a cross-sectional survey conducted between January 2014 and March 2015. A total of 1602 children aged 7–12 years old in the 3rd and 4th grades, from 20 public school located in Porto, Portugal, were invited to participate (Figure 1) [29]. A total of 686 (42.8%) did not present the signed informed consent documents and 58 (3.8%) declined to perform clinical procedures. Among the remaining 858 children (53.6%), 660 (76.9%) had complete nutritional data and were considered for the analysis. From the 660 children who had complete nutritional data, 150 (22.7%) had Na^+^, K^+^ and conductivity measurements of the EBC. Except for overweight/obese children (35.3% versus 22.7%) and asthma diagnosis (16.7% versus 8.5%), no significant differences were observed for age, sex distribution and nutritional data between included and not-included participants (data not shown).

Written consent was obtained from every child’s legal guardian. The Ethics Committee of the University Hospital São João approved the study on 20 September 2013. The study was performed in accordance with Helsinki Declaration. (ARIA 248-13).

### 2.1. Dietary and Diet Quality Assessment

Dietary data were gathered using a single interviewer-administered 24 h food recall questionnaire, completed by the children in accordance with standard procedures. Participants were asked in detail about their food and beverage intakes in the previous 24 h, including brands and quantities, and portion sizes were estimated using a photograph atlas [30].

Total energy intake (kcal) and nutritional data were estimated through the software Food Processor^®^ (ESHA Research, Salem, OR, USA), which encompasses databases of Portuguese nutritional food composition.

Diet quality was assessed through the HEI-2015, a measure used to assess diet quality, specifically to determine which dietary pattern aligns with the Dietary Guidelines for Americans [31]. HEI-2015 is composed of 13 components that add up to a maximum score of 100 points and is divided into two sections: adequacy and moderation. Higher intakes are reflected by higher scores on the nine adequacy components (total fruits, whole fruits, total vegetables, greens and beans, whole grains, dairy, total protein foods, seafood and plant proteins, and fatty acid ratio). The remaining four components are moderation components (refined grains, sodium, added sugars, and saturated fats), with higher scores indicating lower intakes [31]. Because all index components are considered equally important, the HEI components are weighted equally. Some dietary groups have two components, each with a maximum of 5 points; all other elements are worth up to 10 points. Total fruits, whole fruits, total vegetables, total protein foods, greens and beans, seafood, and plant proteins are components that can be scored up to 5 points and these components have 0 points when no foods from the components’ groups are consumed [31]. Whole grains, fatty acid ratio, refined grains, dairy, sodium, added sugars, and saturated fats may go up to a maximum of 10 points and they have a minimum score (= zero) when the minimum of consumption defined by the HEI-2015 is not reached. For more information on the matter, please see for reference Krebs-Smith et al. [31].

Each component of the HEI-2015 is scored on a density basis out of 1000 calories, apart from fatty acids, which is a ratio of unsaturated to saturated fatty acids. Added sugars and saturated fats are displayed as a percentage of total energy intake [31,32].

In terms of arranging the score’s components: total fruits contains 100% fruit juice as well as whole fruits. Meat, poultry, eggs, seafood, nuts, seeds, soy products, and legumes (beans and peas) are elements of total protein foods. Seafood and plant proteins include seafood, nuts, seeds, soy products, and legumes (beans and peas). Legumes (beans and peas) and dark-green vegetables are included in greens and beans. Legumes (beans and peas), dark-green vegetables, and all other vegetables are included in total vegetables [31]. The dairy component includes all milk products such as fluid milk, yogurt, and cheese, as well as fortified soy beverages; saturated fat is counted separately for dairy, meat, poultry, and eggs. When considering nuts, seeds, and soy products, it includes nuts, seeds, and soy products (other than beverages) [31]. The document entitled “Portions and Weights, 2017–2018 Food and Nutrient Database for Dietary Studies-At A Glance” was used to convert dietary information from grams to cups, as data were registered in grams [33].

Intakes between the minimum and maximum standards are proportionately scored. The higher the overall HEI-2015 score, the higher the diet quality and adherence to a healthy eating pattern is considered to be [31,32]. The HEI-2015 was categorized into three groups according to tertile score (1st: ≤54.53; 2nd: >54.53 and ≤65.37; 3rd: >65.37).

#### Supplementation

To assess child nutritional supplementation, a positive response to the question “Has your child taken nutritional supplements (vitamins/minerals) in the last year ?” was used as a determinant.

### 2.2. Anthropometry

Weight was recorded in kilograms using a digital scale (TanitaTM BC-418 Segmental Body Analyzer, Middlesex, United Kingdom), and height was measured in centimeters using a portable stadiometer (cm). BMI was calculated as weight/height^2^ and displayed in kilograms per square meter (kg/m^2^). Participants were categorized into two groups, non-overweight/obese (*p* < 85th) and overweight/obese (*p* ≥ 85th) [34], according to specific age and sex percentiles provided by the US Center for Disease Control and Prevention (US CDC) [35]. The use of the US CDC definition was based on an evaluation of the degree of agreement among various BMI classifications (US CDC, World Health Organization, International Obesity Task Force, and percentage of body fat), with the US CDC showing the highest degree of agreement with all the other classifications (data not presented) [36].

### 2.3. Socio-Economic Data

The number of completed school years was used to calculate the parental education level. It was then divided into three categories: ≤9 years, between ≥10 and ≤12 years, and >12 years, based on the parent with the higher education level, and was used to represent socio-economic status.

### 2.4. Collection and Analysis of the Exhaled Breath Condensate

The Turbo 14 DECCS condenser system (Medivac, Parma, Italy) is a portable device which was used to collect the EBC samples from the children. EBC was collected at the children’s classrooms, from January to April 2014 and October 2014 to March 2015. A saliva trap is incorporated into Turbo 14 DECCS. Samples were obtained by at least 15 min of normal breathing while wearing a nose clip. For each, 800 to 1500 µL of EBC were collected in general, as a volume of at least the 600 µL stipulated as the minimal requirement for a valid sample [37]. To decrease possible sample contamination by environmental air, samples were transferred, in a controlled environment through a laminar flow cabinet, to capped glass tubes after collection and stored at −80 °C till analysis [38].

After EBC sample thawing, Na^+^ EBC was evaluated using the “Compact Na^+^ Meter: B-722-LAQUAtwin” and K^+^ was assessed using the “Compact K^+^ Meter: B-731-LAQUAtwin”. Both incorporate the original HORIBA flat sensor and allow us to obtain accurate measurements of Na^+^ and K^+^ concentrations (mg/L) from just one drop of sample. Conductivity was measured using the “Compact Conductivity Meter: B-771-LAQUAtwin”, in the same conditions. The sensor operates from 5 °C to 40 °C, with an 85% relative humidity maximum, and was calibrated with a 1.41 mS/cm standard solution. It incorporates the original HORIBA flat sensor and allows us to obtain accurate conductivity measurements (mS/cm) from just one drop of sample. Conductivity was measured by the 2-AC bipolar method. The bipolar method measures the current that passes through the solution between a pair of electrodes.

A cut-off from the median was considered for Na^+^ (<39.00 and ≥39.00), K^+^ (<11.00 and ≥11.00), Na^+^/K^+^ ratio (<3.15 and ≥3.15) and conductivity (<0.10 and ≥0.10) to represent lower and higher concentrations values, based on the values obtained from the children’s EBC.

### 2.5. Statistical Analyses

All statistical analyses were performed using the SPSS^®^ statistical package software v27.0 and R studio software. A skewness and kurtosis tests was used to check normality for continuous variables. The characteristics of the participants are presented for the whole sample by sex as percentages for categorical variables, and as median (25th–75th percentile) for non-normal distributed continuous variables, and as mean ± standard deviation (SD) for normal distributed continuous variables. Differences between groups for continuous normally distributed variables were assessed through the Student’s *t* test for independent variables, whereas the Mann–Whitney test was performed for non-normally distributed variables. The chi-square test was used for categorical variables. The associations between HEI-2015 score, both as continuous variable and as tertiles and EBC markers were estimated using logistic regression models (OR, 95% CI). Significant differences were defined with an α-value of less than 5% (*p* < 0.05).

## 3. Results

The characteristics of participants included in the analyses are presented in Table 1.

The mean age of children was 8.67 (±0.75) years, and 48.3% (n = 73) were female. The prevalence of overweight or obese (*p* ≥ 85th) and asthma was 35.8% (n = 54) and 16.6% (n = 25), respectively. We found no significant differences between males and females except for conductivity, with boys [0.12 mS/cm (0.08; 0.24), respectively] presented higher values compared to girls [0.08 mS/cm (0.05; 0.16)].

After making adjustments for asthma, age, sex, total energy intake, BMI, school, nutritional supplementation, and parental education, there was an increased odds of having higher conductivity values (median ≥ 0.10 mS/cm) when children had a higher quality of diet score (OR = 1.04 (95%CI 1.00; 1.08, *p* value = 0.028, Table 2). When HEI-2015 score was divided into tertiles, we found a statistically significant association with conductivity for the third tertile: OR = 4.55 (95% 1.12; 18.45, *p* value = 0.034, Table 2).

## 4. Discussion

Our study found higher odds of having higher levels of conductivity of the exhaled breath condensate when diet had a superior quality score value. However, the present study did not find statistically significant associations between EBC sodium, potassium, or sodium/potassium with diet quality in school-aged children from our sample.

The ionic balance in our body, including the lung, is regulated by extremely efficient mechanisms [39]. Nonetheless, we do not know the magnitude of the influence of diet on these mechanisms.

Conductivity seems to be a marker of the level of ammonium in the EBC [11,40]. A study including healthy individuals and subjects with different respiratory diseases, showed that pH, conductivity, and ammonium were three independent factors that correlated linearly [11]. Moreover, as the correlation between conductivity and ammonium was almost perfect, Dressel et al. proposed that conductivity could be used as a surrogate marker of ammonium content as its measurement is easier to perform compared to standard ammonium detection methods [11].

Both pH and ammonium levels decrease during acute asthma exacerbations in adults [13,41]. Some studies reported that asthmatic children had significantly lower levels of ammonium than the controls [12,42].

In human airway epithelial cell cultures, inflammatory cytokines such as TNF-α and INF-γ reduce glutaminase expression and activity, which is essential to the production of ammonium (a buffer of airway lining fluid acidity) [12,13]. A higher dietary quality, with better nutrient adequacy and quality, may provide several dietary components with antioxidant and anti-inflammatory properties that might decrease the production of pro-inflammatory cytokines [26]. This phenomenon could contribute to normal glutaminase functioning, consequently maintaining the physiologic airway surface liquid acidity by permitting ammonium production. This is reflected in airway EBC conductivity (Figure 2).

Acidification is a common finding in inflamed fluids throughout the body, and it is reasonable to expect the same in the lung in asthma and other inflammatory airway diseases [43]. Some dietary patterns (e.g., the Western diet) have pro-inflammatory effects. In contrast, higher quality dietary patterns, rich in fruits, vegetables, whole grains, and healthy fats, seems to have the opposite effect [26]. Mendes et al. also showed that a higher diversity of vegetables, independently of the amount of consumption, was associated with less airway inflammation and a lower prevalence of asthma [29].

With the previous findings in mind, we may hypothesize that a diet with higher quality may help improve our pulmonary health through its antioxidant and anti-inflammatory properties, counterbalancing a possible abnormal acidity in the airways.

Rama et al. [44] found a bimodal distribution of EBC pH, and a multivariate analysis showed that asthma was significantly associated with a lower EBC pH in the acidic group [β = −0.18 (CI 95%: −0.35, −0.02)]; additionally, the effect of asthma with wheezing in the last 12 months in EBC pH was stronger in the acidic group [β = −0.26 (−0.51, −2.89)] compared with the alkaline group, which showed no association [44]. Based on these previous findings, a sensitivity analysis shown that children in the acidic subgroup and in the third tertile of the HEI-2015 score (higher diet quality) had almost significant results, appearing to be more likely to have higher conductivity levels of the EBC, with an aOR of 5.91 (CI 95%: 1.00, 34.82).

We recognize several limitations in our research. Firstly, the cross-sectional design precludes the establishment of causal relationships between diet quality and the studied EBC markers. However, the same research team completed a detailed collection of data, guaranteeing a relatively unbiased estimate of outcome prevalence. Diet assessment and EBC collection were performed on the same day. Secondly, regarding EBC markers, our study did not take into account the possibility of dilution by water vapor; consequently, individual solute concentrations within airway lining fluid cannot be precisely assessed. However, assessing those values may be important if concentrations differ considerably or are based on solute ratios present in the sample [45]. We considered the Na^+^/K^+^ ratio in our study to overcome this limitation.

Our study has the limitation of having a sample with more overweight/obese and asthmatic children, so our results cannot be generalized to all school-aged children.

Our EBC marker results could be influenced by various biases, including variables that are difficult to control in children outside a clinical setting. We do not have information on the consumption of foods or drinks or exercise the hour before the EBC collection, as these factors can alter the acidity and compound concentrations of the EBC [1,46,47]. Likewise, our study did not consider the presence of gastroesophageal reflux disease, which is a potential modifier of EBC composition [48,49].

In addition, avoiding the contamination of samples from air and saliva may pose a challenge [1], especially in children. However, our study collected EBC during tidal breathing using a nose clip, and the Turbo 14 DECCS incorporated a saliva trap. The nose clip prevents accidental inhalation–exhalation through the nose. When tidal breathing is used for sample collection without resistance, the soft palate is not fully closed and could add air and mediators present in the sinuses and nose to the sample. Using a saliva trap prevents saliva contamination of the samples [50].

Another limitation of this study is the use of one single 24 h recall questionnaire, as multiple recalls are preferred to report an individual’s habitual intake because one single day may not represent the usual diet consumption patterns [51]. This method primarily focuses on short-term intake and does not account for seasonality. Nonetheless, detailed information on the ingredients used in mixed dishes, commercial product brand names, and common size containers was gathered, allowing for a good characterization of dietary intake and consumption. Furthermore, one single 24 h recall questionnaire can estimate the current diet without causing changes in children’s dietary behaviors, in contrast with the time-consuming task of recording diet with the awareness that diet is being assessed [52].

Even though children had to recall all of the foods and beverages they had consumed the day before, a more difficult cognitive task, such as comparing their food intake in the last 24 h to a typical day, was not considered.

Due to limited food knowledge and memory, children’s self-reports of diet are more likely to contain errors, and dietary data collected may suffer from recall bias and indirect reporting [53]. However, to avoid misreporting in children’s dietary consumption, nutritionists and specially trained interviewers conducted 24 h food recall questionnaires with the children using photographs and food models to quantify portion sizes, with the advantage that they have experience probing information from children without suggesting responses [54]. Because assessing food intake is a difficult task, it is easier for children to remember the most recent foods consumed. The 24 h recall may be preferable when determining the typical dietary intake of large groups of subjects [55].

Even though the HEI-2015 is not adapted for Portugal or Portuguese children, this index has the important advantage of being scored on a density basis [31,32]. It employs a less restrictive approach to defining standards for maximum scores, enabling it to be applied to different groups, including children, and evaluating quality over quantity [31]. Moreover, other studies have also used 24 h-reports to assess diet quality [56,57], and Kirkpatrick et al. [57] found that HEI-2015 scores based on 24 h dietary recall data are generally well estimated [57].

Regarding anthropometry, weight classifications were made using BMI. BMI is calculated as weight/height^2^ and does not consider subcutaneous fat, visceral adipose tissue, or body composition fat mass. Therefore, BMI should not be the only factor to take into consideration [58]. Body adiposity could be utilized instead of BMI [59].

Our study also has some strengths. To our knowledge, this study is the first one to evaluate the association between diet quality with selected exhaled breath markers, namely ionic content and conductivity, paving the way for future research on this topic.

As parents may underestimate their children’s overweight/obesity status, we calculated BMI using measured height and weight, avoiding parental self-perceptions of weight categories [60]. Furthermore, we objectively measured bronchodilation by spirometry, combining obtained data with ISAAC self-reported answers to characterize asthma [61].

Dietary patterns, rather than individual dietary components, are being studied to a greater extent nowadays. This is because they may express the effects of interactions between whole food matrixes [62]. The HEI-2015 index also targets food subgroups that are usually lower in diets and have an exceptional nutrient profile, such as dark green vegetables, legumes, and seafood [31]. The Healthy Eating Indexes have the advantage of being continuously updated to match the most recent guidelines [31]. The Healthy Eating Indexes also have the benefit of not requiring any single food to have better scores, but rather allows the researcher to consider the whole diet to characterize diet quality [31] and have a more holistic approach to evaluate diet in a way that accounts for potential interactions between the components.

To better address effects of diet, repeated assessments are required. Biomarkers that confirm the values found in the diet and in the EBC, such as urinary sodium and potassium, would also be useful. As a result, it is critical to conduct prospective studies to evaluate causality and changes over time and determine the cumulative effects of dietary habits and diet quality on lung pathophysiology.

## 5. Conclusions

There seems to be an association between higher diet quality and higher conductivity levels of the exhaled breath condensate in school-aged children from our sample. The association between conductivity and diet may elucidate a new marker that can be used to explore the effects of diet on human respiratory health.

## Figures and Tables

**Figure 1 children-10-00263-f001:**
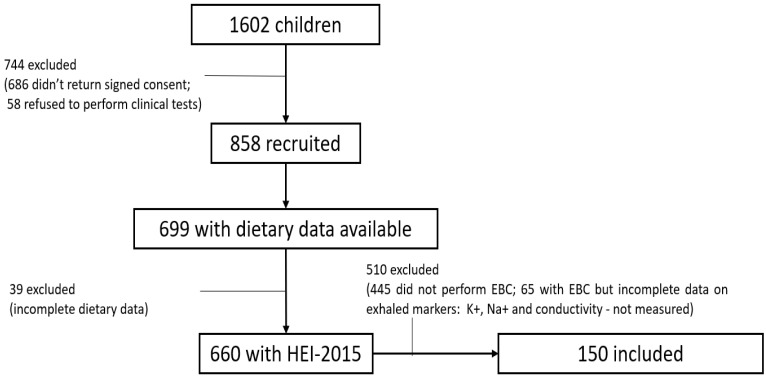
Flow chart of the included participants.

**Figure 2 children-10-00263-f002:**
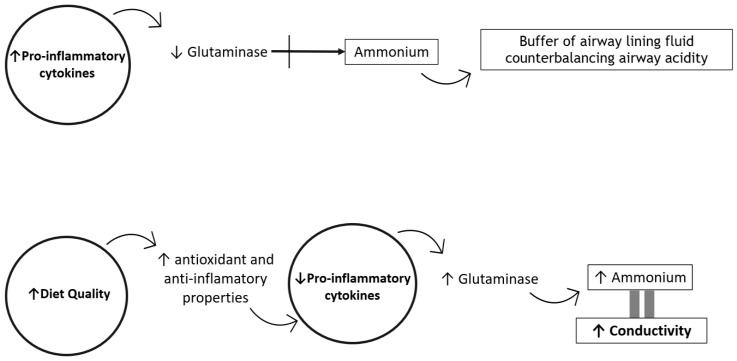
Hypothetical mechanisms by which diet quality may influence EBC conductivity. Pro-inflammatory cytokines decrease glutaminase activity in airway epithelial cells. Glutaminase is responsible for ammonium production, which is considered a buffer of airway acidity. A higher diet quality may provide several dietary components with antioxidant and anti-inflammatory properties that decrease the production of pro-inflammatory cytokines, contributing to normal glutaminase activity and ammonium production, which is reflected in the higher conductivity of the exhaled breath condensate.

**Table 1 children-10-00263-t001:** Summary of participant’s characteristics.

	Total, n = 150	Girls, n = 73 (48.7%)	Boys, n = 77 (51.3.%)	*p* Value
Age (years), mean ± SD	8.67 ± 0.75	8.61 ± 0.71	8.74 ± 0.78	0.283
BMI				0.540
Non-overweight/obese (*p* < 85th)	97 (64.2%)	49 (67.1%)	48 (62.3%)	
Overweight/obese (*p* ≥ 85th)	53 (35.3%)	24 (32.9%)	29 (37.7%)	
HEI-2015 score, mean ± SD	59.94 ± 11.63	60.01 ± 11.46	59.88 ± 11.87	0.943
Carbohydrates, %VET	50.29 ± 6.26	50.94 ± 5.54	49.67 ± 6.85	0.209
Protein, %VET	17.48 ± 3.51	17.69 ± 3.38	17.37 ± 3.64	0.481
Fat %VET	29.26 ± 5.59	28.48 ± 5.65	29.99 ± 5.48	0.097
MONO %VET	10.43 ± 2.81	10.01 ± 2.57	10.82 ± 2.99	0.075
POLI %VET	3.98 ± 1.56	3.80 ± 1.56	4.16 ± 1.55	0.161
SATURATED %VET	9.17 ± 2.94	9.18 ± 3.05	9.17 ± 2.85	0.968
Fiber (g), median (25th–75th)	18.69 (14.75–23.98)	19.70 (15.77–24.43)	18.45 (14.22–23.34)	0.346
Sodium (mg), median (25th–75th)	1978.28 (1598.02–2707.48)	1838.67 (1484.53–2551.02)	2104.24 (1720.35–2867.94)	0.078
Total energy intake (kcal), mean ± SD	2236.03 ± 465.03	2181.69 ± 488.04	2287.54 ± 439.09	0.164
Nutritional Supplementation ^a^, n (%)	22 (16.5%)	11 (17.7%)	11 (15.5%)	0.728
Asthma: Medical diagnosis with asthma symptoms or +BD ^b^	25 (16.7%)	16 (21.9%)	9 (11.7%)	0.093
Parental education ^c^, n (%)				0.984
<9 years	43 (35.5%)	20 (36.4%)	23 (34.8%)	
10–12 years	51 (42.1%)	23 (41.8%)	28 (42.4%)	
>12 years	27 (22.3%)	12 (21.8%)	15 (22.7%)	
Exhaled Breath Markers				
Na^+^, median (25th–75th)	39.00 (28.75–47.25)	40.00 (29.38–47.75)	38.00 (26.50–46.50)	0.657
K^+^, median (25th–75th)	11.00 (4.00–50.50)	8.67 (4.18–49.00)	11.00 (4.00–55.00)	0.523
Conductivity, median (25th–75th)	0.10 (0.06–0.18)	0.08 (0.05–0.16)	0.12 (0.08–0.24)	0.010 *
Na^+^/K^+^ ratio, median (25th–75th)	3.15 (0.67–9.08)	3.60 (0.63–9.80)	2.79 (0.84–8.05)	0.559

Note: * statically significant differences. Abbreviations: HEI: Healthy Eating Index. +BD: Positive Bronchodilation. ^a^ child took nutritional supplement in the previous 12 months; ^b^ at least a 12% and over 200 mL increase in FEV1.; ^c^ number of successfully completed years of formal schooling.

**Table 2 children-10-00263-t002:** Analysis of the association between diet quality with sodium, potassium, and conductivity.

	HEI Score: CrudeModel OR (95% CI)	HEI Score Tertiles: CrudeModel OR (95% CI)		HEI Score: aOR (95% CI)		HEI Score Tertiles: aOR (95% CI)				
	Continuous	>54.53 and ≤65.37	>65.37	Continuous	*p* Value	Reference ≤ 54.53	>54.53 and ≤65.37	*p* Value	>65.37	*p* Value
Na^+^	1.01 (0.98; 1.03)	1.24 (0.55; 2.81)	0.88 (0.41; 1.9)	1.01 (0.98; 1.04)	0.640	1	1.29 (0.36, 4.63)	0.694	1.26 (0.39, 4.11)	0.702
K^+^	0.99 (0.96; 1.02)	1.31 (0.58; 2.99)	0.54 (0.25; 1.19)	0.998 (0.97; 1.03)	0.885	1	1.64 (0.44, 6.05)	0.462	0.78 (0.22, 2.8)	0.704
Na^+^/K^+^ ratio	1.01 (0.98; 1.03)	0.53 (0.23; 1.22)	1.46 (0.67; 3.17)	1.00 (0.97; 1.04)	0.932	1	0.5 (0.13, 1.88)	0.304	1.37 (0.38, 4.98)	0.628
Conductivity	1.02 (1.00; 1.05)	1.74 (0.77; 3.97)	1.95 (0.9; 4.26)	1.04 (1.00; 1.08)	0.037 *	1	1.66 (0.65, 4.27)	0.742	4.55 (1.12, 18.45)	0.034 *

Note: * statically significant differences. Abbreviations: HEI: Healthy Eating Index. aOR: Adjusted OR—age, sex, body mass index categories, nutritional supplementation, school, parent scholarity, total energy intake, and asthma.

## Data Availability

The data that support the findings of this study will be made available by the authors upon reasonable request.

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
