# Peer review of "Diet Quality and Exhaled Breath Condensate Markers in a Sample of School-Aged Children"

_children, 2023, doi:10.3390/children10020263_

Round 1

Reviewer 1 Report

Dear Authors,

Thank you for the excellent work. The manuscript is very well written and worth publishing. In my opinion, the article would be of great interest to readers.

I have the most technical comments.

Please indicate the mean age of the study participants in the abstract (±SD).

Next, in my opinion, the term "conductivity" and its associations with respiratory system health characteristics should be better explained in the Introduction section. Also, the assessment of the conductivity and the cut-offs (<0.10 and ≥0.10) should be better explained in the Methods section.

Next, I wonder if the adequacy of water/fluids intake and body hydration were taken into account when testing the associations between diet quality and the EBC.

Please provide study limitations if there are such.

Reviewer 2 Report

Title could be misleading because your results are reflecting a subgroup of school-age children, that are more overweight and have more asthma than general pediatric population in your country.

Introduction has many data on irrelevant aspects for current study, like gluten-free studies or nitrosative pathways and many others. Rows 64-75 don't seem to belong to the theoretical construct that generated current study. please provide only current state-of-the-art knowledge related to evaluated biomarkers.

The aim of the study is not obvious. Please rephrase why should such a study be performed and how clinicians or patients could profit out of these efforts.

Material and Methods - please elaborate on inclusion/exclusion criteria. Please clarify discrepancies mentioned in rows 94-97 and if so how this subgroup compares to general pediatric population in Portugal.

Methodology query - why did you elaborate in-extenso on asthma criteria? Is this part of an asthma research or is an independent study regarding general population?

Please elaborate on methodology... Where was this process performed? Children were evaluated in a single center or device was portable and was moved to children's school?   

Discussion section has many non-related comments to current study and discordancy with referenced papers. Paper 24 has incomplete citation.

Rows 297-300 and 311-316 raise significant concerns about soundness of results.

Conclusions should be revised because you can not extrapolate to whole group of school-age children this subgroup of rather overweight and asthmatic children and because it is not evident the added-value of your research for pediatric patients. 

Round 2

Reviewer 2 Report

Authors just added a nuance in title in order to avoid confrontation with the really serious selection-bias from methodology ["Diet quality and exhaled breath condensate markers in a sample of school-aged children" instead of "Diet quality and exhaled breath condensate markers in school-aged children"].

This inserted element "in a sample" is an attempt to bypass a serious question posed in initial review : "Material and Methods - please elaborate on inclusion/exclusion criteria. Please clarify discrepancies mentioned in rows 94-97 and if so how this subgroup compares to general pediatric population in Portugal."

As one can see no explanations were provided and issue was nor clarified in my opinion. I think that national or at least local [from Porto metropolitan area] data on obese/overweight are needed.

There is no statement in introduction that a predetermined goal of study is to evaluate asthma patients or overweight patients as special subgroups... Why any other comorbid issues were not explored in similar exhaustive way? Like allergic, renal, or cardiovascular status, immune response or vaccine coverage? It seems beyond the scope of current paper, that is focusing on school age children not on "patients", to explore in detail any given disease.

As previously noted, in first review, this paper it seems to be generated by an asthma study. In order to avoid such a distracting perspective on present findings it is advisable to delete the entire section between 178-198 rows.

Please revise data in following section, because data are not concordant.

Rows 256-258 are stating:

"When HEI-2015 score was divided 256 into tertiles, it was observed a statistically significant association with conductivity for the 257 third tertile: OR = 4.55 (95% 1.12; 18.45, p-value=0.034, Table 3)."

and in table 3 are these elements

"Conductivity 1.02 (1.00;1.05) 1.68 (0.74,3.85) 1.89 (0.86,4.13) 1.04 (1.00;1.08) 0.037* 1 1.22 (0.32,4.62) 0.765 4.32 (1.05,17.74) 0.042"

and Rows 325-327 are stating :

"had almost significant results, appearing to be more likely to have higher conductivity levels of the EBC, with an aOR of 5,91 (CI 95%: 1.00, 34.82"
